# Sodium channel-inhibiting drugs and cancer-specific survival: a population-based study of electronic primary care data

Caroline Fairhurst,[1] Tim Doran,[1] Fabiola Martin,[2] Ian Watt,[1] Martin Bland,[1] William J Brackenbury  [3]

[1]Department of Health Sciences, University of York, York, UK
[2]School of Public Health, University of Queensland, Brisbane, Queensland, Australia
[3]York Biomedical Research Institute, Department of Biology, University of York, York, UK

**Correspondence to**
Dr William J Brackenbury;
william.brackenbury@york.ac.uk

## ABSTRACT

**Objectives** Antiepileptic and antiarrhythmic drugs inhibit voltage-gated sodium (Na$^+$) channels (VGSCs), and preclinical studies show that these medications reduce tumour growth, invasion and metastasis. We investigated the association between VGSC inhibitor use and survival in patients with breast, bowel and prostate cancer.

**Design** Retrospective cohort study.

**Setting** Individual electronic primary healthcare records extracted from the Clinical Practice Research Datalink.

**Participants** Records for 132 996 patients with a diagnosis of breast, bowel or prostate cancer.

**Outcome measures** Adjusted Cox proportional hazards regression was used to analyse cancer-specific survival associated with exposure to VGSC inhibitors. Exposure to non-VGSC-inhibiting antiepileptic medication and other non-VGSC blockers were also considered. Drug exposure was treated as a time-varying covariate to account for immortal time bias.

**Results** During 1 002 225 person-years of follow-up, there were 42 037 cancer-specific deaths. 53 724 (40.4%) patients with cancer had at least one prescription for a VGSC inhibitor of interest. Increased risk of cancer-specific mortality was associated with exposure to this group of drugs (HR 1.59, 95% CI 1.56 to 1.63, p<0.001). This applied to VGSC-inhibiting tricyclic antidepressants (HR 1.61, 95% CI 1.50 to 1.65, p<0.001), local anaesthetics (HR 1.49, 95% CI 1.43 to 1.55, p<0.001) and anticonvulsants (HR 1.40, 95% CI 1.34 to 1.48, p<0.001) and persisted in sensitivity analyses. In contrast, exposure to VGSC-inhibiting class 1c and 1d antiarrhythmics was associated with significantly improved cancer-specific survival (HR 0.75, 95% CI 0.64 to 0.88, p<0.001 and HR 0.54, 95% CI 0.33 to 0.88, p=0.01, respectively).

**Conclusions** Association between VGSC inhibitor use and mortality in patients with cancer varies according to indication. Exposure to VGSC-inhibiting antiarrhythmics, but not anticonvulsants, supports findings from preclinical data, with improved survival. However, additional confounding factors may underlie these associations, highlighting the need for further study.

## INTRODUCTION

Metastatic disease is the leading cause of death from solid tumours,[1] and there is an

### STRENGTHS AND LIMITATIONS OF THIS STUDY

⇒ Primary care research data with large sample size and statistical power.
⇒ Drug exposure is treated as a time-varying covariate to account for immortal time bias.
⇒ No direct information on metastasis as an outcome.
⇒ Drug exposure data are based on prescriptions.

enduring need to identify new antimetastatic targets and therapies.[2] One approach is to repurpose existing drugs used in the management of other conditions. In particular, ion channel blockers have been proposed as novel agents to treat cancer, including metastatic disease.[3] However, no such agent has yet been progressed through to clinical use.

Voltage-gated sodium (Na$^+$) channels (VGSCs) are expressed on electrically excitable cells including neurons and muscle cells, where they regulate action potential firing.[4] VGSC-inhibiting drugs are prescribed for a range of excitability-related conditions, including epilepsy, pain and cardiac arrhythmia.[5 6] VGSCs are also widely expressed on malignant cells from a range of cancers, where they regulate Na$^+$ handling, pH buffering and the plasma membrane potential, promoting proliferation, migration, invasion and metastasis.[7–12] Numerous preclinical studies have shown that VGSC-inhibiting medications can reduce tumour growth, invasion and metastasis.[13–21] Although some antiepileptic drugs have been tested in clinical trials,[22 23] their effect on VGSC activity in patient tumours has not been investigated. Several observational cohort studies have shown reduced cancer incidence[24 25] and risk of recurrence[26–28] in patients prescribed VGSC-inhibiting medications. In contrast, we have previously reported that exposure to VGSC-inhibiting medication was associated

with reduced overall survival in cancer patients in a retrospective analysis.[29] However, we were unable to control for epilepsy diagnosis, which is independently associated with increased all-cause mortality.[30] In this study, we conducted a retrospective cohort study using primary care data from the Clinical Practice Research Datalink (CPRD) in order to test the hypothesis that exposure to VGSC inhibitors prolongs cancer-specific survival. We controlled for epilepsy diagnosis and timing of exposure to VGSC-inhibiting drugs and considered other antiepileptic medications.

## METHODS

### Patient data

The study protocol has been published previously.[31] Several additional analyses were performed as detailed below. Primary care records for patients with a first diagnosis of any cancer between 2001 and 2011 and aged 25 years or over at diagnosis were obtained from the CPRD GOLD and Aurum databases. CPRD contains anonymised individual patient data on morbidity, mortality, prescribing, treatment and referrals collected from primary care practices in England. Data were extracted in August 2019. Within this dataset, we identified patients with a recorded medical code for breast, bowel or prostate cancer (hereafter referred to as the index cancers). The role of VGSCs has been extensively studied in these three types of cancer, and they are among the most common in the UK.[12 31] Prescription data were interrogated to identify patients with a recorded prescription for VGSC-inhibiting medications (including anticonvulsants, local anaesthetics, antiarrhythmics and certain tricyclic antidepressants; online supplemental table 1). We also identified patients with a recorded prescription for non-VGSC-inhibiting anticonvulsants (eg, gabapentinoids, benzodiazepines) and medications targeting other (non-voltage-gated) $Na^+$ channels (eg, the epithelial $Na^+$ channel), at any time (online supplemental table 1). We searched diagnostic codes to identify patients with a recorded VGSC inhibitor indication (epilepsy, cardiac arrhythmia, amyotrophic lateral sclerosis (ALS) and neuropathic pain).[5 6]

### Statistical analysis

Time-dependent Cox proportional hazards regression was used to analyse survival time from cancer diagnosis associated with exposure to the medication group of interest, and all models were adjusted for type of index cancer, sex and age at diagnosis ($age+age^2$), unless otherwise stated. Right censoring occurred if the patient died of any other cause, was still alive at the point the data were extracted or the patient transferred out of a CPRD general practitioner (GP) practice.

To account for potential immortal time bias[32] in patients whose prescriptions only begin after their cancer diagnosis, drug exposure status was considered as a time-dependent covariate in the following three ways:

Scenario 1: all person-time of follow-up from diagnosis to death/censor was classified as exposed for patients who have at least one prescription of interest before their diagnosis; while for those who only have prescriptions after their diagnosis, their survival time was classified as unexposed between diagnosis and date of first prescription and as exposed thereafter.

Scenario 2: person-time was considered as unexposed until the date of the first prescription and as exposed thereafter for patients whose prescriptions either: (1) start before diagnosis and extend after or (2) start after diagnosis. This differs from scenario 1 in that, for patients whose first and last prescriptions are before their cancer diagnosis, their survival time is treated as exposed in scenario 1 and unexposed in this scenario.

Scenario 3: person-time was considered as unexposed until the date of the first prescription following the date of cancer diagnosis and as exposed thereafter. This differs from scenario 2 in that, for patients whose prescriptions of interest start before diagnosis and extend after, the time between diagnosis and the first prescription after diagnosis is considered exposed in scenario 2 and unexposed in this scenario.

In all scenarios, all person-time of follow-up for patients who have never had a recorded prescription of interest was classified as unexposed. A depiction of these scenarios is presented in the published protocol.[31]

Multivariable-adjusted HRs are presented with a 95% CI and p value. Analyses were conducted in Stata V.15,[33] using two-sided statistical tests at the 5% significance level.

Survival graphs were produced using the Simon-Makuch method, which is an alternative to Kaplan-Meier that appropriately accounts for the time-varying covariate of exposure.[34]

### Patient characteristics

Patient characteristics, stratified by 'ever' and 'never' exposure to a VGSC-inhibiting medication, are summarised using mean and SD for continuous data and count and percentage for categorical variables, and compared using a t-test or $\chi^2$ test as appropriate. Amide and ester local anaesthetic injections were not included within the definition of 'exposed', due to their short-term use and transient effect.

Characteristics of the 'ever' exposed group stratified by timing of drug exposure relative to their cancer diagnosis (before only, before and after, and after only) are also presented, including length of drug exposure and most commonly prescribed drug class. Extent of drug exposure was estimated by calculating the time between the first and last recorded prescription, plus a number of weeks (the average interval between all prescriptions) to account for the time patients were assumed to be taking their final recorded prescription. Based on this, patients were classified into short (< 6 months) or long (≥ 6 months) exposure groups. Those who had two or more prescriptions relating to one of the VGSC-inhibiting drugs within 2 years before the date of the cancer

diagnosis, including at least one within 6 months before, were classified as having recent (to cancer diagnosis) exposure. Alternative medications were summarised for patients with a recorded diagnostic code for an indication for a VGSC-inhibiting drug (epilepsy, neuropathic pain, cardiac arrhythmia, ALS) who did not have a recorded prescription for a VGSC-inhibiting drug.

### Primary analysis

The primary analysis investigated cancer-specific mortality (any cancer as the underlying cause) associated with drug exposure, treated as a time-varying covariate according to the three scenarios described above, using adjusted Cox proportional hazards regression models.

### Sensitivity analyses

The primary analyses were repeated with the Cox models additionally adjusted for: ethnicity, body mass index (BMI), physical activity, smoking status, alcohol consumption, Charlson Comorbidity Index (CCI) score, presence of an indication for VGSC-inhibiting medication and area-level social deprivation, using the Index of Multiple Deprivation (IMD) in twentiles (1=least deprived to 20=most deprived) based on patient postcode (2010). In further sensitivity analyses, missing values for the confounding factors (previously included in a 'not recorded' category) were imputed using multiple imputation and the analysis models rerun.

In addition, the primary analysis was repeated using competing-risks regression, according to the method of Fine and Gray and implemented using the stcrreg command in Stata,[35] with death by any other cause but cancer as the competing risk, and also after introducing a lag such that patients were not considered as exposed until 3 months after drug use. This excludes prescriptions shortly before death and, therefore, minimises potential reverse causation.[36]

### Secondary analyses

The primary analyses were repeated stratified by index cancer diagnosis (not adjusting for index cancer, and also removing sex as a covariate for prostate cancer analysis as all patients were male, and for breast cancer as nearly all patients were female) and comparing time to: (1) death from index cancer (underlying or contributory cause); (2) death from any cancer (underlying or contributory cause) and (3) all-cause mortality. The primary analyses were also repeated by including in the 'ever' exposed group, in turn, only patients who had: (1) ever used; (2) had recent exposure (according to definition as above) to or (3) whose most commonly prescribed VGSC inhibitor (not including local anaesthetic injections) was tricyclic antidepressants, anticonvulsants and antiarrhythmics. Other drug classes were not considered due to insufficient numbers in these groups.

### Vaughan Williams classification of antiarrhythmics

We repeated the primary analyses considering exposure to VGSC-inhibiting antiarrhythmics subdivided according to the updated Vaughan Williams classification (online supplemental table 1).[37 38] Exposure to different classes of antiarrhythmic medications was assessed depending on whether the patient's use of the drugs was defined as: (1) ever use, (2) recent use or (3) their most common VGSC inhibitor prescription.

### Amide or ester local anaesthetic injections

We repeated the time-dependent analysis (scenario 3 only since 1 and 2 are not applicable here) including only those patients whose VGSC-inhibiting drug prescriptions were solely for amide or ester local anaesthetic injections (online supplemental table 1) following their diagnosis in the exposed group, since there is evidence that local anaesthetics used perioperatively can be associated with reduced tumour recurrence.[12]

### Non-VGSC-targeting antiepileptic medication

We repeated the primary analyses considering exposure to non-VGSC-targeting antiepileptic medications, and blockers of other (non voltage-gated) $Na^+$ channels (online supplemental table 1).

### Patient and public involvement

None.

## RESULTS
### Population characteristics

The CPRD dataset contained records for 515 987 patients from 1057 GP practices, including 132 996 (25.8%) patients with a diagnostic code for breast (n=59 528), prostate (n=50 601) or bowel (n=22 867) cancer recorded during at least one of their GP consultations. Of the 132 996 index patients with cancer, 79 164 (59.5%) had at least one prescription, at any time, for a specified VGSC-inhibiting drug; tricyclic antidepressant was the most commonly prescribed VGSC-inhibiting drug group for the majority of exposed patients (n=33 905, 42.8%), followed by amide local anaesthetics (n=30 091, 38.0%). For one-third of these 79 164 patients (n=25 440, 32.1%), their only exposure to a VGSC-inhibiting drug was to amide or ester local anaesthetics. These patients were classified as unexposed for most of the described analyses, due to the short-term exposure, so 53 724 (40.4%) patients were observed to have had at least some exposure to a VGSC inhibitor of interest, before and/or after cancer diagnosis, and 79 272 (59.6%) were not (online supplemental table 2). Stratified by index cancer, the proportion of 'ever' exposed patients was: breast 59.5%, bowel 54.7% and prostate 61.7%.

Between the 'ever' and 'never' exposure groups, formal comparisons indicated statistically significant differences in all observed characteristics, even where differences were very small such as in the CCI (mean 6.1 in the 'ever' exposed group and 5.9 in the unexposed group), which is likely to be an artefact of the large sample size (online supplemental table 2). On visual inspection, the two

exposure groups appear similar for most patient characteristics, including age, but there was a notable imbalance in sex, with a greater proportion of females in the 'ever' exposed group than in the unexposed group. There were expected differences in the proportions of patients with an indication for treatment with a VGSC inhibitor; for example, 3.6% of the 'ever' exposed group had a diagnosis of epilepsy, compared with 0.6% of the unexposed group.

Within the 'ever' exposed group, 14 157 patients (26.4%) only had prescriptions of interest dated before a cancer diagnosis, 17 264 (32.1%) had prescriptions dated both before and after diagnosis, and 22 303 (41.5%) only had prescriptions dated after diagnosis (online supplemental table 3). For patients who initiated VGSC inhibitors after their cancer diagnosis, the mean interval between diagnosis and first recorded prescription was 4.0 years (SD 3.5, median 3.0, range 1 day to 18.2 years).

For the subset of patients with a recorded diagnosis of an indication for VGSC-inhibiting medication in their medical records who did not have a recorded prescription for a VGSC-inhibiting drug (n=16 048), the most common prescriptions were for angiotensin-converting enzyme inhibitors (727 736 prescriptions among 9887 (61.6%) patients), lipid-regulating drugs (647 200 prescriptions among 8099 (50.5% patients), antiplatelet drugs (622 772 prescriptions among 10 602 (66.1%) patients), beta-adrenoceptor blocking drugs (515 888 prescriptions among 7798 (48.6%) patients) and voltage-gated calcium channel blockers (503 847 prescriptions among 8044 (50.1%) patients). These proportions were very similar for the subset of patients with a recorded diagnosis of an indication for VGSC-inhibiting medication in their medical records who did have a recorded prescription for a VGSC-inhibiting drug (n=18 744), except that a slightly higher proportion of these patients had a prescription for a beta-blocker (54.9%).

The maximum follow-up from diagnosis was 18.6 years (median 7.9 years). During 1 002 225 person-years of follow-up, there were 66 960 deaths from any cause (online supplemental table 4). A similar proportion of deaths from any cause were recorded in the data for the two groups ('ever' exposed 48.4%, unexposed 51.6%), and of deaths with any cancer listed as the underlying cause (primary outcome, total n=42 037; 'ever' exposed 29.7%, unexposed 32.9%) or as at least a contributory cause (n=32 725; 'ever' exposed 34.6%, unexposed 38.5%) (online supplemental table 4).

### Primary, sensitivity and secondary analyses

The main text focuses on results from analyses relating to scenario 3, as this most closely matches the design of relevant preclinical studies,[13 14] but all results are presented in the tables. In the primary analysis, we considered the relationship between all VGSC inhibitors (excluding local anaesthetics) and cancer-specific survival across all three index cancer types (breast, bowel and prostate) combined. Exposure to VGSC inhibitors was associated

**Table 1** Estimates of the relationship between exposure to VGSC inhibitors and cancer specific mortality—primary and sensitivity analyses

| Cancer-specific mortality (underlying cause) | HR (95% CI) | P value |
|---|---|---|
| Primary analysis | | |
| Scenario 1 | 1.33 (1.31 to 1.36) | <0.001 |
| Scenario 2 | 1.31 (1.28 to 1.34) | <0.001 |
| Scenario 3 | 1.59 (1.56 to 1.63) | <0.001 |
| Sensitivity analyses 1* | | |
| Scenario 1 | 1.42 (1.39 to 1.45) | <0.001 |
| Scenario 2 | 1.38 (1.34 to 1.41) | <0.001 |
| Scenario 3 | 1.65 (1.62 to 1.69) | <0.001 |
| Sensitivity analyses 2† | | |
| Scenario 1 | 1.34 (1.26 to 1.43) | <0.001 |
| Scenario 2 | 1.31 (1.22 to 1.41) | <0.001 |
| Scenario 3 | 1.60 (1.49 to 1.72) | <0.001 |
| Sensitivity analyses 3‡ | | |
| Scenario 1 | 1.34 (1.26 to 1.43) | <0.001 |
| Scenario 2 | 1.35 (1.25 to 1.45) | <0.001 |
| Scenario 3 | 1.65 (1.53 to 1.78) | <0.001 |
| Sensitivity analyses 4§ | | |
| Scenario 1 | 1.20 (1.18 to 1.23) | <0.001 |
| Scenario 2 | 1.17 (1.14 to 1.20) | <0.001 |
| Scenario 3 | 1.37 (1.34 to 1.41) | <0.001 |

*Primary analyses additionally adjusted for ethnicity, BMI, physical activity, smoking status, alcohol consumption, CCI score, IMD score and presence of: epilepsy; cardiac arrhythmias; ALS; neuropathic pain/painful neuropathy.
†Sensitivity analyses one repeated after unknown values of ethnicity, BMI, physical activity, smoking status, IMD and alcohol consumption imputed using multiple imputation.
‡Competing-risks regression using stcrreg command in Stata adjusting for exposure group, type of cancer, sex, age and age$^2$, with death by any other cause but cancer as the competing risk.
§Primary analysis repeated after introducing a 3-month lag to exposure.
ALS, amyotrophic lateral sclerosis; BMI, body mass index; CCI, Charlson Comorbidity Index; IMD, Index of Multiple Deprivation; VGSC, voltage-gated sodium (Na+) channels.

with a statistically significant increased risk of death from cancer (HR 1.59, 95% CI 1.56 to 1.63, p<0.001; table 1; figure 1). The HR increased in the sensitivity analysis additionally adjusted for ethnicity, BMI, physical activity, smoking status, alcohol consumption, IMD, CCI score and presence of a VGSC-inhibitor indication (1.65, 95% CI 1.62 to 1.69) and in the competing-risks analysis (1.65, 95% CI 1.53 to 1.78), but was similar after missing covariate data were imputed although with a wider CI (1.60, 95% CI 1.49 to 1.72). A smaller but still significant effect was observed in the analysis that utilised a lag of 3 months to discount drug use shortly before death (HR 1.37, 95% CI 1.34 to 1.41, p<0.001; table 1).

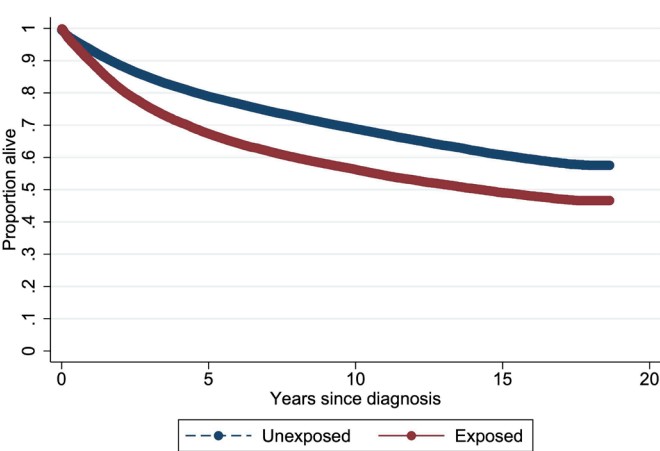

**Figure 1** Simon-Makuch survival curve for unexposed patients with cancer and those ever exposed to VGSC-inhibiting drugs in scenario 3. VGSC, voltage-gated sodium (Na+) channels.

In secondary analyses, we stratified by cancer type; there was a statistically significantly (p<0.001) increased mortality rate associated with exposure to VGSC-inhibiting medication across all three cancers, HR (95% CI) for: breast 1.49 (1.43 to 1.54); prostate 1.65 (1.60 to 1.71) and bowel 1.64 (1.57 to 1.71) (table 2). There was a similar relationship for the outcomes of time to death from specific index cancer (HR 1.58, 95% CI 1.55 to 1.62), cancer as an underlying or contributory cause (1.56, 95% CI 1.53 to 1.60) and all-cause mortality (1.50, 95% CI 1.48 to 1.53) (table 2).

### VGSC-inhibiting anticonvulsants and tricyclic antidepressants

Among patients with exposure to anticonvulsants (ever use n=6391), VGSC inhibitor use was associated with significantly increased risk of death from cancer (HR 1.40, 95% CI 1.34 to 1.48, p<0.001; online supplemental figure 1A; table 3). A higher HR was observed among those for whom anticonvulsants were the most frequent prescription for a VGSC inhibitor (1.62, 95% CI 1.53 to 1.72), but lower for recent use (1.11, 95% CI 1.02 to 1.21). Among patients with exposure to tricyclic antidepressants (ever use n=42 715), VGSC inhibitor use was similarly associated with significantly increased risk of death from cancer (HR 1.61, 95% CI 1.5 to 1.65, p<0.001; online supplemental figure 1B; table 3); again, a higher HR was associated with tricyclic antidepressants being the most frequent prescription for a VGSC inhibitor (1.67, 95% CI .63 to 1.71), but lower (and non-statistically significant) for recent use (0.98, 95% CI 0.93 to 1.04, p=0.59).

A total of 12 140 patients received VGSC-inhibiting drug prescriptions solely in the form of amide or ester local anaesthetic injections following their cancer diagnosis, of which 3656 (30.1%) died with (any) cancer as the underlying cause. Exposure to these injections was associated with a statistically significantly increased risk of death from any cancer (HR 1.49, 95% CI 1.43 to 1.55, p<0.001).

**Table 2** Estimates of the relationship between exposure to VGSC inhibitors and mortality—secondary analyses

| Secondary analyses | HR (95% CI) | P value |
|---|---|---|
| Primary analyses by type of cancer | | |
| Breast | | |
| Scenario 1 | 1.27 (1.23 to 1.32) | <0.001 |
| Scenario 2 | 1.22 (1.18 to 1.27) | <0.001 |
| Scenario 3 | 1.49 (1.43 to 1.54) | <0.001 |
| Prostate | | |
| Scenario 1 | 1.38 (1.33 to 1.42) | <0.001 |
| Scenario 2 | 1.42 (1.37 to 1.47) | <0.001 |
| Scenario 3 | 1.65 (1.60 to 1.71) | <0.001 |
| Bowel | | |
| Scenario 1 | 1.34 (1.29 to 1.40) | <0.001 |
| Scenario 2 | 1.26 (1.21 to 1.32) | <0.001 |
| Scenario 3 | 1.64 (1.57 to 1.71) | <0.001 |
| Death from index cancer (underlying or contributory cause) | | |
| Scenario 1 | 1.33 (1.31 to 1.36) | <0.001 |
| Scenario 2 | 1.30 (1.27 to 1.32) | <0.001 |
| Scenario 3 | 1.58 (1.55 to 1.62) | <0.001 |
| Cancer-specific mortality (underlying or contributory cause) | | |
| Scenario 1 | 1.33 (1.31 to 1.36) | <0.001 |
| Scenario 2 | 1.29 (1.27 to 1.32) | <0.001 |
| Scenario 3 | 1.56 (1.53 to 1.60) | <0.001 |
| All-cause mortality | | |
| Scenario 1 | 1.34 (1.32 to 1.36) | <0.001 |
| Scenario 2 | 1.28 (1.26 to 1.30) | <0.001 |
| Scenario 3 | 1.50 (1.48 to 1.53) | <0.001 |

### Classes 1–3 antiarrhythmics

In contrast to the VGSC-inhibiting anticonvulsants and tricyclic antidepressants, exposure to VGSC-inhibiting antiarrhythmic drugs was associated with decreased risk of cancer-specific mortality (recent use HR 0.92, 95% CI 0.86 to 0.99, p=0.03) or no difference (online supplemental figure 1C; table 3). In exploratory analyses, these drugs were separated into their Vaughan Williams classes (table 4, online supplemental figure 2A–D).[37 38] Exposure to class 1a antiarrhythmic drugs (n=188) had no impact on cancer-specific survival (ever use HR 1.05, 95% CI 0.76 to 1.46, p=0.77; online supplemental figure 2A; table 4). Exposure to Class 1b drugs (n=1088), some of which are also indicated as anticonvulsants (eg, phenytoin), was associated with significantly reduced cancer-specific survival (ever use HR 2.06, 95% CI 1.88 to 2.26, p<0.001; online supplemental figure 2B; table 4). In contrast, exposure to class 1c drugs (n=860) was associated with significantly improved cancer-specific survival (ever use HR 0.75, 95% CI 0.64 to 0.88, p<0.001; online supplemental figure 2C; table 4). The class 1d drug ranolazine (n=165) was associated with significantly improved cancer-specific

Table 3   Estimates of the relationship between exposure to VGSC-inhibiting drugs, subdivided by type and cancer-specific mortality

| VGSC inhibitor drug group | Exposed* (n=53 724), n (%) | HR (95% CI) P value Scenario 1 | HR (95% CI) P value Scenario 2 | HR (95% CI) P value Scenario 3 |
|---|---|---|---|---|
| **Ever use** | | | | |
| Antiarrhythmic | 15 538 (28.9) | 0.91 (0.88 to 0.95) <0.001 | 0.84 (0.80 to 0.87) <0.001 | 1.02 (0.98 to 1.06) 0.34 |
| Anticonvulsant | 6391 (11.9) | 1.19 (1.14 to 1.24) <0.001 | 1.17 (1.12 to 1.23) <0.001 | 1.40 (1.34 to 1.48) <0.001 |
| Tricyclic antidepressant | 42 715 (79.5) | 1.32 (1.29 to 1.35) <0.001 | 1.30 (1.27 to 1.33) <0.001 | 1.61 (1.57 to 1.65) <0.001 |
| **Recent use** | | | | |
| Antiarrhythmic | 2807 (5.2) | 0.95 (0.89 to 1.01) 0.12 | 0.87 (0.81 to 0.93) <0.001 | 0.92 (0.86 to 0.99) 0.03 |
| Anticonvulsant | 1656 (3.1) | 1.14 (1.05 to 1.24) <0.001 | 1.05 (0.96 to 1.15) 0.27 | 1.11 (1.02 to 1.21) 0.02 |
| Tricyclic antidepressant | 5408 (10.1) | 1.01 (0.96 to 1.06) 0.76 | 0.90 (0.85 to 0.95) <0.001 | 0.98 (0.93 to 1.04) 0.59 |
| **Most common VGSC inhibitor prescription** | | | | |
| Antiarrhythmic | 11 032 (20.5) | 0.94 (0.91 to 0.98) <0.001 | 0.87 (0.83 to 0.91) <0.001 | 1.00 (0.95 to 1.05) 0.94 |
| Anticonvulsant | 4062 (7.6) | 1.41 (1.34 to 1.48) <0.001 | 1.45 (1.36 to 1.53) <0.001 | 1.62 (1.53 to 1.72) <0.001 |
| Tricyclic antidepressant | 38 600 (71.9) | 1.36 (1.33 to 1.39) <0.001 | 1.37 (1.33 to 1.40) <0.001 | 1.67 (1.63 to 1.71) <0.001 |

*Figures in this column relate to the number of patients recorded as having at least some follow-up time considered as exposed to the drug class of interest in scenario 1 for each definition (ever use, recent use, most common), as a percentage of the whole 'ever' exposed group. The number of patients with any person-time of follow-up considered as exposed for each drug class will be lower in scenario 2 and fewer still in scenario 3.
VGSC, voltage-gated sodium (Na+) channel.

survival (ever use HR 0.54, 95% CI 0.33 to 0.88, p=0.01; online supplemental figure 2D; table 4). However, class 2 drugs (beta blockers; n=11 643) were not associated with altered cancer-specific survival (ever use HR 0.99, 95% CI 0.94 to 1.04, p=0.70; table 4). Finally, class 3 drugs (which are also K[+] channel blockers; n=3532) also were not associated with altered cancer-specific survival, (ever use HR 1.06, 95% CI 0.98 to 1.13, p=0.14; table 4).

### Non-VGSC-targeting antiepileptic medications and other Na$^+$ channel blockers

To investigate whether the reduced cancer-specific survival of patients exposed to VGSC-inhibiting anticonvulsants is attributable to their Na$^+$ current-inhibiting action, we considered the impact of two other drug groups: (1) anticonvulsants that do not target VGSCs and (2) drugs that target other types of Na$^+$ channels, independent of VGSCs. One-third (n=46 017, 34.6%) of patients had a prescription for a non-VGSC-targeting anticonvulsant, and 7% (n=9256) for a non-VGSC-targeting Na$^+$ channel blocker (online supplemental table 1). For both drug groups, there was a higher proportion of deaths (from any cause) among those exposed than among those not exposed, and this was true when cancer was considered

among the causes of death (online supplemental table 4). Among those who died, patients exposed to a non-VGSC-targeting antiepileptic medication were more likely to die with any cancer as an underlying cause than unexposed patients (71.1% vs 57.7%); whereas patients exposed to a non-VGSC-targeting Na$^+$ channel blocker were less likely (51.7% vs 64.1%). Exposure to both drug groups was associated with increased risk of cancer-specific mortality (HR 4.60, 95% CI 4.51 to 4.70, p<0.001 for non-VGSC-inhibiting anticonvulsants; and 1.42, 95% CI 1.35 to 1.49, p<0.001 for non-VGSC-inhibiting Na$^+$ channel blockers; online supplemental figure 3A,B; table 5). Findings are presented by drug class in online supplemental table 5.

### DISCUSSION

This study shows that exposure to VGSC-inhibiting drugs (anticonvulsants, local anaesthetics and tricyclic antidepressants) in patients with breast, bowel and prostate cancer is associated with a statistically significant increased risk of death from cancer. This risk is elevated for patients who were exposed to this class of medication before, as well as after, their cancer diagnosis. In addition,

**Table 4** Estimates of the relationship between exposure to antiarrhythmic drugs, subdivided by Vaughan Williams classification, and cancer-specific mortality

| Vaughan Williams drug groups | Exposed* (n=53 724) n (%) | HR (95% CI) P value Scenario 1 | HR (95% CI) P value Scenario 2 | HR (95% CI) P value Scenario 3 |
|---|---|---|---|---|
| **Ever use** | | | | |
| 1a Fast VGSC block, K⁺ channel block | 188 (0.3) | 0.94 (0.73 to 1.21) 0.64 | 0.88 (0.64 to 1.22) 0.45 | 1.05 (0.76 to 1.46) 0.77 |
| 1b† VGSC block, fast association/ disassociation | 1088 (2.0) | 1.82 (1.67 to 1.99) <0.001 | 1.84 (1.68 to 2.02) <0.001 | 2.06 (1.88 to 2.26) <0.001 |
| 1c VGSC block, slow association/ disassociation | 860 (1.6) | 0.73 (0.64 to 0.84) <0.001 | 0.67 (0.57 to 0.78) <0.001 | 0.75 (0.64 to 0.88) <0.001 |
| 1d Persistent current block | 165 (0.3) | 0.41 (0.25 to 0.68) <0.001 | 0.42 (0.26 to 0.68) <0.001 | 0.54 (0.33 to 0.88) 0.01 |
| 2 Beta adrenergic block | 11 643 (21.7) | 0.87 (0.84 to 0.91) <0.001 | 0.79 (0.75 to 0.83) <0.001 | 0.99 (0.94 to 1.04) 0.70 |
| 3 K⁺ channel block | 3532 (6.6) | 0.98 (0.92 to 1.04) 0.50 | 0.92 (0.85 to 0.98) 0.02 | 1.06 (0.98 to 1.13) 0.14 |
| **Recent use** | | | | |
| 1a | 45 (0.1) | 1.17 (0.73 to 1.88) 0.52 | 1.00 (0.59 to 1.69) 1.00 | 1.06 (0.63 to 1.79) 0.83 |
| 1b* | 429 (0.8) | 1.22 (1.05 to 1.42) 0.01 | 1.15 (0.98 to 1.34) 0.10 | 1.21 (1.03 to 1.42) 0.02 |
| 1c | 298 (0.6) | 0.82 (0.66 to 1.01) 0.06 | 0.80 (0.65 to 0.99) 0.04 | 0.84 (0.68 to 1.04) 0.11 |
| 1d | 4 (0.0) | – | – | – |
| 2 | 1752 (3.3) | 0.92 (0.84 to 1.00) 0.06 | 0.84 (0.77 to 0.92) <0.001 | 0.89 (0.81 to 0.98) 0.01 |
| 3 | 738 (1.4) | 1.03 (0.92 to 1.16) 0.59 | 0.94 (0.83 to 1.07) 0.37 | 1.01 (0.89 to 1.14) 0.91 |
| **Most common VGSC inhibitor prescription** | | | | |
| 1a | 107 (0.2) | 1.06 (0.77 to 1.45) 0.71 | 0.97 (0.62 to 1.52) 0.89 | 1.10 (0.70 to 1.73) 0.67 |
| 1b* | 756 (1.4) | 1.95 (1.76 to 2.16) <0.001 | 1.97 (1.76 to 2.19) <0.001 | 2.16 (1.94 to 2.41) <0.001 |
| 1c | 632 (1.2) | 0.76 (0.65 to 0.89) <0.001 | 0.71 (0.59 to 0.86) <0.001 | 0.78 (0.65 to 0.94) 0.01 |
| 1d | 126 (0.2) | 0.43 (0.24 to 0.78) 0.01 | 0.44 (0.24 to 0.79) 0.01 | 0.57 (0.31 to 1.02) 0.06 |
| 2 | 8025 (14.9) | 0.92 (0.88 to 0.96) <0.001 | 0.84 (0.79 to 0.89) <0.001 | 1.01 (0.95 to 1.07) 0.84 |
| 3 | 2786 (5.2) | 1.03 (0.96 to 1.10) 0.46 | 0.96 (0.89 to 1.04) 0.34 | 1.08 (1.00 to 1.17) 0.06 |

*Figures in this column relate to the number of patients recorded as having at least some follow-up time considered as exposed to the drug group of interest in scenario 1 for each definition (ever use, recent use, most common), as a percentage of the whole 'ever' exposed group. The number of patients with any person-time of follow-up considered as exposed for each drug group will be lower in scenario 2 and fewer still in scenario 3.
†Excluding lidocaine, which is commonly prescribed as a local anaesthetic.
VGSC, voltage-gated sodium (Na+) channel.

both non-VGSC-targeting anticonvulsants and non-VGSC-targeting Na⁺ channel blockers are associated with significantly increased risk of death from cancer. Notably, the risk of death from cancer is approximately two times higher for non-VGSC-targeting vs VGSC-inhibiting anticonvulsants. In contrast, VGSC-inhibiting antiarrhythmic

**Table 5** Estimates of the relationship between exposure to non-VGSC-inhibiting anticonvulsants, non-VGSC-inhibiting Na⁺ channel blockers and cancer-specific mortality

| Cancer-specific mortality (underlying cause) | Non-VGSC-inhibiting anticonvulsant | | Non-VGSC-inhibiting Na$^+$ channel blocker | |
|---|---|---|---|---|
| | HR (95% CI) | P value | HR (95% CI) | P value |
| Scenario 1 | 2.98 (2.92 to 3.04) | <0.001 | 1.25 (1.21 to 1.30) | <0.001 |
| Scenario 2 | 3.60 (3.53 to 3.68) | <0.001 | 1.32 (1.26 to 1.39) | <0.001 |
| Scenario 3 | 4.60 (4.51 to 4.70) | <0.001 | 1.42 (1.35 to 1.49) | <0.001 |

VGSC, voltage-gated sodium (Na+) channel.

medications display a different pattern, and are associated with moderately improved cancer-specific survival. When subdivided according to the updated Vaughan Williams classification, class 1c and 1d VGSC inhibitors (which have slow receptor association/disassociation, producing persistent current block) are associated with significantly improved cancer-specific survival in several scenarios.

### Strengths and weaknesses of the study
The study uses data from the CPRD, the largest prospectively collected primary care database in the UK containing information on causes of death, comorbidities and drug exposure based on prescription data.[39 40] We studied cancer-specific mortality in addition to overall mortality, and we controlled for other potentially confounding life-limiting indications for which VGSC-inhibiting medications are prescribed.[5 6 31 41] A key limitation of observational studies of drug effects on survival is immortal time bias, where patients in the exposed group can enter an 'immortal' period in the follow-up time between index diagnosis and first prescription of the drug under study.[42] We implemented a person-time approach to control for this issue, where exposure status was considered as a time-dependent covariate.[31 42] However, this adjustment did not alter the overall conclusions. We also conducted analyses that added a lag of 3 months to exposure to minimise issues of reverse causation; again, conclusions were unchanged.

There are several important limitations to the study. First, GP records, including diagnostic codes, covariate data and prescription information, may be incomplete or contain errors. Additionally, a prescription record does not account for non-adherence, and so exposure to the drugs of interest is inferred. Second, although the dataset was linked to causes of death, it was not linked to secondary care databases, including the National Cancer Data Repository,[43] and so we did not have access to information on cancer stage, progression or treatment. Third, although we were able to identify those patients with cancer who had a diagnostic code for a confounding life-limiting indication, for example epilepsy, we had limited information on the severity of the conditions, which is linked to both medication use and survival. It is possible that additional uncontrolled confounding factors in the population may underlie the associations, for example, cardiovascular complications,[44 45] underscoring a key

problem with such retrospective cohort studies. We also did not measure metastasis directly, so further work is required to establish why patients with cancer exposed to these medications have altered survival.

### Comparison with other studies
Our findings partially agree with our previous study showing that exposure to VGSC-inhibiting medications is associated with reduced overall survival of patients with cancer.[29 46] Refinements to the design of the current study, including adjustment for epilepsy diagnosis, and analysis of cancer-specific survival in addition to overall survival,[31] did not alter this conclusion. However, subdividing VGSC inhibitors according to their primary indication revealed positive associations between exposure to antiarrhythmics (in particular class 1c and 1d drugs) and cancer-specific survival. In addition, the current study showed for the first time that the negative association between anticonvulsant exposure and cancer-specific survival was greater for non-VGSC-targeting anticonvulsants than for VGSC-inhibiting anticonvulsants. A number of preclinical studies indicate that VGSC-inhibiting medications reduce survival, proliferation, migration, invasion and metastasis of cancer cells.[13–15 47–50] These would support the hypothesis that such drugs may have value as anti-metastatic agents. In addition, several clinical studies have shown valproate, another VGSC blocker, to have antitumour activity.[23 51–53] However, this may, at least partially, be as a result of its action as a histone deacetylase inhibitor.[15 54]

### Implications for clinical practice
The disagreement between the preclinical observations and the primary care data presented here raises the possibility that any beneficial effect of VGSC-inhibiting medications on cancer progression may be masked by larger effects in the population. We previously postulated that estimation of a positive association may be affected by confounding by indication.[29] VGSC-inhibiting medications are indicated primarily for epilepsy, but are also prescribed for other life-limiting conditions, including cardiac arrhythmias, ALS and neuropathic pain/painful neuropathy.[5 6 31 41] Epilepsy patients have an elevated risk of death from all causes, including cancer, compared with the general population (standardised mortality ratio >2.2),[30 55 56] possibly due to a poorer general health and/or social status.[44 57 58] Adjustment for comorbidities

and social deprivation had no effect on the relationship between exposure and reduced survival. In addition, several VGSC-inhibiting antiepileptic drugs, including carbamazepine and phenytoin, can induce activity of the hepatic cytochrome P450 isoenzyme system, which in turn metabolises certain chemotherapeutic agents, including camptothecin analogues, methotrexate, taxanes, teniposide and vinca alkaloids.[59 60] Some VGSC inhibitors, including phenytoin, have also been shown to impact on immune function.[61]Alterations in bioavailability and efficacy of chemotherapeutic agents in the presence of VGSC inhibitors, as well as potential interactions with other treatments, should be studied further.

The observation that non-VGSC-targeting anticonvulsants were associated with worse survival than VGSC-inhibiting anticonvulsants raises the possibility that VGSC inhibition may indeed be beneficial in this cohort of patients with cancer, thus indirectly supporting the preclinical hypothesis.[9] Moreover, the improved cancer-specific survival of patients exposed to class 1c and 1d antiarrhythmics, which preferentially target the persistent $Na^+$ current that is responsible for VGSC-dependent metastatic behaviour in preclinical models,[11 14 37] further supports the notion that inhibition of these channels may be beneficial in the clinical setting. However, we cannot exclude the possibility that other confounders may exist between patients within these subgroups, for example, epilepsy or cardiac arrhythmia severity.

## CONCLUSIONS

The unique positive association between antiarrhythmic drug prescriptions and improved survival may point to a specific beneficial effect of certain VGSC inhibitors with this indication, for example, ranolazine,[14 21] and warrants further investigation. These results should be replicated in a study with robust cancer stage data, and an appropriately designed and controlled prospective clinical trial to establish the effect of VGSC inhibition on tumour progression. Such a trial would separate possible uncontrolled confounding from cancer-specific mortality and could also exploit emerging novel pathophysiological biomarkers of disease progression, for example, circulating tumour DNA and $^{23}$Na-MRI.

**Correction notice** This article has been corrected since it was first published. The author's name 'William J Brackenbury' and his affiliation have been updated.

**Acknowledgements** The authors acknowledge the Medical Research Council and the Wellcome Trust for funding this study.

**Contributors** CF, TD and WJB had the original idea for this study. CF conducted the analysis under supervision of TD and WJB. CF, TD and WJB wrote the draft of the manuscript. IW, FM and MB contributed to the development of the idea, the study design, interpretation of the findings and revising the manuscript. All authors approved the final submitted version of the manuscript.

**Funding** This work was supported by the Medical Research Council (G1000508) and Wellcome Trust through the Centre for Future Health at the University of York (award number not available).

**Competing interests** None declared.

**Patient and public involvement** Patients and/or the public were not involved in the design, or conduct, or reporting, or dissemination plans of this research.

**Patient consent for publication** Not applicable.

**Ethics approval** This study was performed following ethics approval from the Department of Biology Ethics Committee, University of York (WB201909). GPs do not seek individual patient consent when they share deidentified data with the CPRD (CPRD policy here: https://www.cprd.com/public). The study was performed in accordance with the Declaration of Helsinki.

**Provenance and peer review** Not commissioned; externally peer reviewed.

**Data availability statement** Data are available on reasonable request. The datasets used and/or analysed during the current study are available from the corresponding author on reasonable request.

**ORCID iD**
William J Brackenbury http://orcid.org/0000-0001-6882-3351

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
