## [Reviewer comments · BMJ Open]

ARTICLE DETAILS

TITLE (PROVISIONAL)	Sodium channel-inhibiting drugs and cancer-specific survival: a population-based study of electronic primary care data
AUTHORS	Fairhurst, Caroline; Martin, Fabiola; Watt, Ian; Bland, Martin; Doran, Tim; Brackenbury, William

VERSION 1 – REVIEW

REVIEWER	Aktas, Hatice Gumushan Harran Universitesi
REVIEW RETURNED	15-Jul-2022

GENERAL COMMENTS	In this study, the authors investigated the association between VGSC inhibitor use and survival in breast, bowel, and prostate cancer patients based on the CPRD database as a retrospective cohort study. All parts of the manuscript were written appropriately. Additionally, this study's results, strengths, and limitations were discussed in detail. The results of this study may provide important data for new experimental and clinical investigations.
--

REVIEWER	Chawla , Pooja A. ISF College of Pharmacy, Pharmaceutical Chemistry
REVIEW RETURNED	11-Sep-2022

GENERAL COMMENTS	1. The language of manuscript should be in concise Manner and please revise the grammatical and language error. 2. Add the justification of choosing only breast, bowel, and prostate cancer patient's.
--

REVIEWER	Altamura, Concetta University of Bari, Dept. of Precision and Regenerative Medicine
REVIEW RETURNED	30-Dec-2022

GENERAL COMMENTS	In this study, the authors use data from the CPRD to study cancer-specific mortality after the treatment of patients with VGSC blockers. In summary, they demonstrated that exposure to VGSC-inhibiting drugs like local anaesthetics, antidepressant and anticonvulsants is associated with a statistically significant increased risk of death in patients with breast, bowel and prostate cancer. In contrast, Class 1c and 1d VGSC-inhibiting antiarrhythmics are associated with significantly improved cancer-specific survival. Furthermore, both non-VGSC-targeting anticonvulsants and non-VGSC-targeting Na ⁺ channel blockers are associated with significantly increased risk of death from cancer.
--

	The paper analyzed the main aspects of the proposed topic, it is clear and well written.
--	--

VERSION 1 – AUTHOR RESPONSE

Reviewer 1:

In this study, the authors investigated the association between VGSC inhibitor use and survival in breast, bowel, and prostate cancer patients based on the CPRD database as a retrospective cohort study. All parts of the manuscript were written appropriately. Additionally, this study's results, strengths, and limitations were discussed in detail. The results of this study may provide important data for new experimental and clinical investigations.

We thank the Reviewer for their positive assessment of the manuscript.

Reviewer 2:

1. The language of manuscript should be in concise Manner and please revise the grammatical and language error.

We have re-proofread the manuscript to check for grammatical and language errors.

2. Add the justification of choosing only breast, bowel, and prostate cancer patient's.

A justification is provided in the Methods section on Patient data: "The role of VGSCs has been extensively studied in these three types of cancer, and they are among the most common in the UK (12,31)."

Reviewer 3:

In this study, the authors use data from the CPRD to study cancer-specific mortality after the treatment of patients with VGSC blockers.

In summary, they demonstrated that exposure to VGSC-inhibiting drugs like local anaesthetics, antidepressant and anticonvulsants is associated with a statistically significant increased risk of death in patients with breast,

bowel and prostate cancer. In contrast, Class 1c and 1d VGSC-inhibiting antiarrhythmics are associated with significantly improved cancer-specific survival.

Furthermore, both non-VGSC-targeting anticonvulsants and non- VGSC-targeting Na⁺ channel blockers are associated with significantly increased risk of death from cancer.

The paper analyzed the main aspects of the proposed topic, it is clear and well written.

We thank the Reviewer for their positive assessment of the manuscript.